# High-Efficiency Mono-Cyclopentadienyl Titanium and Rare-Earth Metal Catalysts for the Production of Syndiotactic Polystyrene

**DOI:** 10.3390/molecules28196792

**Published:** 2023-09-25

**Authors:** Bo Wen, Hongfan Hu, Di Kang, Chenggong Sang, Guoliang Mao, Shixuan Xin

**Affiliations:** 1Provincial Key Laboratory of Polyolefin New Materials, College of Chemistry & Chemical Engineering, The Northeast Petroleum University, Daqing 163318, China; 2PetroChina Petrochemical Research Institute, PetroChina Company Limited, Beijing 102206, China

**Keywords:** metallocenes, titanocene catalysts, RE metals, styrene, syndiotactic polystyrene

## Abstract

Syndiotactic polystyrene (SPS) refers to a type of thermoplastic material with phenyl substituents that are alternately chirally attached on both sides of an aliphatic macromolecular main chain. Owing to its excellent physical and mechanical properties, as well as its chemical stability, high transparency, and electrical insulation characteristics, SPS is used in a wide variety of technical fields. SPS is commonly produced via the stereoselective transition metal-catalyzed coordination polymerization method mediated by stereospecific catalysts, which consists of anionic mono-cyclopentadienyl derivative η^5^-coordinated single active metal centers (referred to as “mono-Cp’-M”), with active center metals involving Group 4 transition metals (with an emphasis on titanium) and rare-earth (RE) metals of the periodic table. In this context, the use of mono-cyclopentadienyl titanocene (mono-Cp’Ti) catalysts and mono-cyclopentadienyl rare-earth metal (mono-Cp’RE) metallocene catalysts for the syndiospecific polymerization of styrene is discussed. The effects of the mono-cyclopentadienyl ligand structure, cationic active metal types, and cocatalysts on the activity and syndiospecificity of mono-Cp’ metallocene catalysts are briefly surveyed.

## 1. Introduction

Among the vast majority of synthetic resin products, the predominant commodity plastic materials with saturated carbon backbones are polyethylene (PE), polypropylene (PP), polyvinyl chloride (PVC), polystyrene (PS), polyacrylonitrile (PAN), and other copolymers, as well as products containing heteroatoms (O, N, S) and aromatic rings in the main chain, such as polyamide (PA), polyimide (PI), polycarbonate (PC), polyester (PES), polyurethane (PU), polysulfone (PSU), etc., which appear in manufactured goods and daily necessities [1]. SPS resin is a high-end thermoplastic material that possesses indispensable, high application values, but it has relatively high production technology barriers, resulting in only a few producers dominating the international market [2].

SPS is an important thermoplastic material not only because it has excellent physical and mechanical properties but also because of its high levels of thermostability chemical stability, transparency, dimensional stability, and low density (ca. 1.10 g/cm^3^), which have led to the high demand for SPS in diverse technical applications.

On the other hand, styrene can be copolymerized with other available monomers to obtain copolymers with the expected properties. For example, styrene may be copolymerized with acrylonitrile to improve the stress cracking resistance of the produced polymer (SCN) [3]; copolymerized with acrylonitrile and butadiene to produce a commodity copolymer (ABS), which has improved heat and chemical resistance and greatly expanded the processing window [4]; and copolymerized with butyl acrylate and acrylonitrile, resulting in a terpolymer (AAS) that greatly increases the material’s lifetime to about 8–10 times that of conventional polystyrene [5].

Polystyrene (PS) homopolymers and copolymers are widely used in the field of transportation, playing an indispensable role especially with the rise of new energy vehicles [6]. In the medical field, a range of medical devices use PS materials [7]. PS and its composite materials are widely used in the production of electronic equipment, home appliances, and other products [8]. PS materials still play a role in the traditional construction field [9] and are also widely used in the field of food and cosmetics packaging [10].

Styrene can be polymerized via all known polymerization technologies, including widely used anionic polymerization, free radical polymerization, and coordination polymerization [11,12,13,14]. Research on the polymerization of controlled stereospecific PS first began in the 1950s, when Professor Natta obtained isotactic polystyrene (IPS) using the TiCl_4_/AlR_3_ catalytic system [15,16]. With the discovery of the MAO cocatalyst’s effects on single active central metal-organic catalysts [17] and the later discovery that arylboron cocatalysts can largely improve the performance of metallocene catalytic systems [18], the study of stereospecific styrene polymerization has experienced rapid advances in recent decades. Researchers can regulate the activity and stereoselectivity of catalytic systems by varying the ligand structure, the type of central metal, the type of cocatalyst, or a combination of the two or three [19,20,21,22], which largely facilitates the development of the scope and dimensions of SPS catalysts.

Due to configurational attachment (or chirality) differences in the phenyl substituents on the saturated linear aliphatic main chain, polystyrene can be divided into three main structural categories, namely isotactic polystyrene (IPS), with all phenyl substituents distributed on the same side of the polymer main chain; syndiotactic polystyrene (SPS), with phenyl substituents alternately distributed on both sides of the polymer backbone; and atactic polystyrene (APS), with phenyl substituents randomly attached on both sides of the polymer backbone in equal proportions [23]. A Fischer projection of the three polystyrene structures is shown in Figure 1.

Atactic polystyrene (APS) exhibits good electrical insulation properties and excellent processability. As a thermoplastic polymeric material, APS is widely used in packaging, daily necessities, and other industries. As early as the 1930s, I. G. Farbenindustrie (Liverkusen, Germany) and Dow (Midland, MI, USA) developed and produced a limited amount of APS [1], and over time, metallocene catalysts have also been cited in the development of APS [24].

Highly stereospecific IPS and SPS were developed after the emergence of relevant stereospecific catalysts. IPS has a high degree of crystallinity and high melting temperature (up to 240 °C), but due to its slow crystallization rate, its product performance is affected by processing procedures. In the 1950s, Natta first successfully used the Ziegler–Natta (Z-N) catalytic system to synthesize fully isotactic polystyrene (IPS), a process that was soon scaled to industrial production [25].

SPS has an even higher melting temperature (about 270 °C), and at the same time, SPS has a high degree of crystallinity, a fast crystallization rate, excellent chemical resistance to acid and alkali, good dimensional stability, low moisture absorption, and excellent processing characteristics at very low melt viscosity. In the 1980s, Ishihara successfully synthesized SPS using a homogeneous titanium catalytic system activated with methylaluminoxane (MAO), and the resulting polymer was carefully characterized [26]. As a promising engineering plastic with diverse application potential, SPS has attracted the interest of many researchers from industry and academia for developing new high-efficiency catalytic systems.

## 2. Properties and Application of Syndiotactic Polystyrene (SPS)

### 2.1. SPS Condensed Matter Structure and Properties

SPS has four stable crystal forms, namely α, β, γ, and δ crystal forms. α and β crystal forms have molecular chains adopted an inverse planar conformation with an equivalent period of 0.51 nm, while γ and δ crystal forms possess a spiral conformation with an equivalent period of 0.77 nm [27]. The densities of the four crystal forms are: α, 1.034 g/cm^3^; β, 1.078 g/cm^3^; γ, 1.10 g/cm^3^; and δ, 0.977 g/cm^3^. SPS in different crystal forms has different applications [28]. SPS in different crystal forms can interconvert under the induction of temperature and/or solvents [29,30,31].

The special spatially oriented configuration of syndiotactic polystyrene is the fundamental determinant of the physical-mechanical characteristics of SPS. Therefore, the stereoregularity of the main chain is closely related to the physical properties of SPS. The sequence structure and degree of syndiotacticity of PS can be calculated by the resonance absorption intensity of the chemical shifts in the characteristic ^13^C and ^1^H NMR spectra. Using the resonance absorption intensities of the methylene carbon on the main chain and the phenyl substituent’s *ipso*-carbon, the stereoregularity of the polymer can be accurately calculated. In general, methylene carbon appears at high field in the 42–47 ppm region, while the phenyl substituent’s *ipso*-carbon is distributed at low field in the 145–147 ppm region. It is believed that the *isotactic*-pentad (*mmmm, or meso-pentad where the adjacent two phenyl substituents have the same chiral configuration*) occurs in the lowest C-13 NMR resonance field, whereas the *syndiotactic*-pentad (*rrrr, or racemic-pentad where the adjacent two phenyl substituents have the opposite chiral configuration*) occurs in the highest C-13 NMR resonance field. The following Equations (1) and (2) can be used to calculate the stereoregularity of PS polymeric materials [32,33,34].
***Pmmmm* = *w*(1 − *u*)3/(*u* + *w*)**(1)
***Prrrr* = *u*(1 − *w*)3/(*u* + *w*)**(2)
where *u = Pr{r|m} = 1 − Pr{m|m}, w = Pr{m|r} = 1 − Pr{r|r}*. *Pmmmm* and *Prrrr* are the proportions of isotactic *mmmm*-pentads and syndiotactic *rrrr*-pentads, respectively; and *u* and *w* are the probabilities of the *r* and *m* configurations, respectively.

In addition to the property advantages of SPS materials, they also exhibit drawbacks such as brittleness and temperature- and/or solvent-induced crystal form interconversion. Copolymerization of styrene with other monomers may be a solution to overcoming some of the physical property drawbacks of SPS [35,36].

In more than half a century, the research and development of SPS catalytic systems have gone through several stages, from simple transition metal complexes (MXn) to dicyclopentadienyl metal (metallocene) complexes, and further to highly stereoselective and highly active mono-Cp’ metallocene compounds. Metallocenes, especially mono-Cp’ metallocene catalytic systems, have the characteristics of homogeneous polymerization processes, tolerating high polymerization temperature and producing SPS with higher syndiotacticity. Thus, mono-Cp’ metallocene catalytic systems have become a focal point for the research and development of special SPS materials [37], along with extensive studies of cocatalytic systems. The use of methylaluminoxane (MAO), steric bulky arylborane, and boric salts with adjustable steric bulkiness and Lewis acidity have greatly improved the activity and stereoselectivity of catalytic systems [38,39] and expanded the application fields of SPS.

### 2.2. Applications of SPS Materials

SPS in automotive applications includes, but is not limited to, high-temperature, high-humidity, and highly compacted areas under the hood (many components, limited space, limited airflow); miniaturized components in the car (GPS devices, DVD players, etc.); precision insulated connectors between wiring terminals; SPS tracking resistant connectors, terminals, and fuse holders in hybrid vehicles; the vehicle’s 600-V power system integration architecture; automotive or electronic printed circuit board connectors; and automotive electronic control module connectors.

The inherent low electrical power loss of SPS, along with the ability of molten-state low-viscosity SPS to easily fill thin-walled, complex mold cavities, has led to significant growth in SPS applications in electronic components, such as embedded antennas and various interconnect devices that rely on consistent electrical performance. The adhesion of SPS to other SPS substrates is sufficient, which is an advantage in the design and manufacture of secondary components.

Using the characteristics of heat and chemical resistance, and excellent dimensional stability to ensure the tight tolerance required for precision extruded parts, SPS is widely used in precision machinery and household appliances. For example, liquid-filled transformer components require long-term contact with mineral oil used for coolants as well as heat resistance. Due to the excellent hydrolytic stability of SPS, it is used in many parts related to water treatment, including pump housings, impellers, and water heaters. The chlorine resistance of SPS also makes it suitable for components exposed to domestic water. Resistant to steam, pressurized hot water, ethylene oxide, and even gamma radiation, SPS is ideal for manufacturing equipment that requires sterilization. Due to its good bio-compatibility (following the ISO 10993 measurement standard), SPS is used in non-fluid/tissue contact medical devices. Heating, ventilation, and air conditioning (HVAC) components employ SPS’s hydrolysis and heat resistance, as well as its chemical resistance, to withstand the corrosive flue gases generated. The electric meter base takes advantage of SPS’s excellent electrical insulation properties and high heat resistance. A wide range of electrical components utilize SPS for hydrolysis, heat, and dimensional stability, including coffeemakers, waste compactors, humidifiers, and washer/dryer components.

The following context only deals with the role of mono-Cp’Ti and mono-Cp’RE metallocene catalysts in the syndiospecific polymerization of styrene.

## 3. Mono-Cp’ Metallocene Catalysts

### 3.1. Mono-Cp’ Metallocene Catalyst Characteristics

Metallocene catalysts, also known as cyclopentadienyl transition metal complex catalysts, refer to metal complex precursors composed of ligands containing at least one η^5^-cyclopentadienyl derivative and a transition metal (or RE metal). Compared with traditional Ziegler–Natta olefin polymerization catalysts, metallocene catalysts have only one single active center and the metal atoms in the active centers are generally confined in a space-restricted environment, allowing only monomers to coordinate to the active site of the catalyst and accurately controlling the monomer coordination orientation, thereby controlling the relative molecular weight, molecular weight distribution, stereoregularity, and crystalline structure of the polymers [40,41,42,43].

The active center of a mono-Cp’ metallocene catalyst is generally a positively charged transition metal (or RE metal). After an ancillary alkyl ligand heterolytic is cleaved off from the central metal in the form of a negative ion under the action of a cocatalyst, the mono-Cp’ metallocene molecule becomes an electron-deficient metal cation. The olefin molecule is a weakly basic Lewis base (LB), which easily π-coordinates to the cationic metal center in the confined coordination sphere, realizing the insertion of the first coordinated monomer double-bond into metal-alkyl bonds, completing chain initiation, followed by consecutive monomer coordination/insertion chain propagation steps [44].

In addition to cationizing the central metals in mono-Cp’ metallocene catalytic systems, cocatalysts can also stabilize the negative ions by delocalizing the negative charge over the bulky anionic framework.

### 3.2. Mono-Cp’ Titanocene Catalysts

Group 4 elements play a predominant role in olefin polymerization catalysis. The earliest styrene syndiospecific polymerization used an MAO-activated titanium catalytic system [26]. Compared with Zr-based catalysts, the Ti-based catalysts with identical structural features exhibited significantly higher catalytic activity and stereoselectivity [45]. Compared with the classic mono-Cp’-Ti (η^5^-C_5_H_5_)TiCl_3_ and (η^5^-C_5_Me_5_)TiCl_3_ catalytic systems, the bis(cyclopentadienyl)titanium catalytic systems, (η^5^-C_5_H_5_)_2_TiCl_2_ and (η^5^-C_5_Me_5_)_2_TiCl_2_ [46], exhibited very low activity and low syndioselectivity. Interestingly, the bis(cyclopentadienyl) zirconium and hafnium analogs, [(η^5^-C_5_H_5_)_2_ZriCl_2_ and (η^5^-C_5_H_5_)_2_HfCl_2_], showed very poor catalytic activity and produced only atactic polystyrene (APS). At the same time, comparison of the mono-Cp’ titanocene catalyst with the Cp’_2_TiCl_2_ catalyst in styrene syndiospecific polymerization found that the yield of SPS was independent of the σ-coordinated ligand steric hindrance in both catalytic systems. It is worth noting that when increasing the proportion of organoaluminum scavenger in the (η^5^-C_5_H_5_)TiCl_3_ and (η^5^-C_5_Me_5_)TiCl_3_ catalytic systems, the activity was significantly enhanced [46], suggesting that alkylaluminium was also acting as an alkylation reagent.

Kaminsky reported a mono-Cp’ titanocene trifluoride complex (**2**) and compared it with a classic mono-Cp’ titanocene trichloride complex (**1**) (Figure 2). In the temperature range of 10–70 °C, **2** was more active than **1**, while the melting point (*T_m_*) of PS obtained by **2** was higher than that of **1**, suggesting that the PS obtained from **2** possessed higher syndiotacticity in its macromolecular structure. When the cyclopentadienyl ring (Cp) was fully substituted with methyl groups (complexes **3** and **4**), the activity of trifluoride **4** was also higher than that of the trichloride analog (**3**). Interestingly, by changing the Al:Ti ratio, the activities of **3** and **4** were altered accordingly. When the Al:Ti ratio was 300, **4** had higher activity, and when the Al:Ti ratio increased to 900, **3** had higher activity, suggesting that the trichloride metallocene complexes were less reactive with the alkylaluminum reagent, and thus demanding a larger excess of Lewis acidic alkylaluminum to reach the same level of cationization compared with the fluorinated analogs. It was also found that using pentamethylcyclopentadienyl (Cp*) ligand could improve the styrene polymerization activity compared to simple Cp, and that mono-Cp* titanocene systems produced SPS with higher molecular weight and stereoregularity. This indicated that the mono-Cp*Ti catalytic system had higher stability than its simple mono-Cp-Ti counterpart, thus maintaining a high proportion of stable active catalysts and offering a longer catalytic lifetime to retard catalytic deactivation processes [47].

Rausch reported a series of mono-Cp’ titanocene alkoxy complexes (Figure 2, **5**–**9**). When activated with MAO, the complexes exhibited distinct activities toward styrene syndiospecific polymerization. Complex **6** showed the highest catalytic activity and stereoselectivity at 75 °C, while the activity of complex **9** was much lower than that of **6.** On the other hand, **9** produced SPS with the highest molecular weight. It was speculated that the bulky and electron-withdrawing trimethylsilyl substituent enhanced the electron density of the cyclopentadienyl group in complex **9** and increased steric hindrance, resulting in lower polymerization activity but SPS with higher stereospecificity and molecular weight [48].

Liu reported a series of mono-alkoxy mono-Cp titanocene [CpTiCl_2_ (OR)] complexes (Figure 2, **10**–**15**). In previous studies, the researchers found that a single alkoxy (OR) group could increase the activity of the classic trichloride titanocene catalyst. The steric hindrance of the R group could affect the polymerization activity of the mono-Cp titanocene complexes. Among the six mono-Cp titanocenes, complex **10** had the highest activity, and the styrene monomer’s conversion rate increased with rising reaction temperature, but the time required for complete monomer conversion was increased, resulting in decreased catalytic efficiency at higher temperatures accompanied with lower syndiotacticity, suggesting that the catalytic reactivity and stereoselectivity were highly monomer concentration-dependent. Further kinetic studies on the catalytic system should provide credible insight to the catalytic system. Reaction temperature also played a undeniable role in stereo-control; altering the reaction temperature from 70 °C to 90 °C reduced the syndiotacticity of SPS from 92.9% to 72.6% [49].

Huang et al. reported two η^1^-alkene-substituted cyclopentadienyl mono-Cp’ titanocene complexes (**16**–**17**) and compared them with corresponding saturate n-alkyl-substituted mono-Cp’ titanocene analogs (**18**–**19**). The results showed that the activities of complexes **17** and **19** were higher than those of **16** and **18**, indicating that increasing the chain length of the substituent could enhance the catalytic activity. Surprisingly, the two complexes containing unsaturated substituents (**16** and **17**) exhibited opposite activity and selectivity, with the complex containing the allyl group (**16**) having the lowest activity, and the complex containing the butenyl group showing the highest activity. In general, the introduction of bulky substituents in the Cp ring reduced the catalytic activity. It was believed that the coordination and insertion of the monomer may not have been affected by the misalignment of the butenyl group with the active center or weak coordination of the butenyl double bond [50].

Nomura et al. introduced a series of mono-Cp’ titanocenes with electron-donating groups on the Cp ring and alkoxy auxilliary ligands (**20**–**27**). The results showed that complexes containing the alkoxy group were generally more active toward styrene polymerization than the classic trichloride Cp’TiCl_3_ derivative. The catalytic activity decreased as the number of substituents increased on the Cp ring on the one hand, while introducing electron-donating substituents on the Cp ring seemed to reduce the chain transfer rate relative to chain growth on the other hand, thus producing higher molecular weight SPS. The catalytic activity also varied with the substituents of the σ-coordinated aryloxy group, and complex **26** showed the highest catalytic activity. Thus, not only the substituents on the ring but also those on the ancillary donor ligands (alkoxy, amide, aniline) played a non-negligible role in influencing the polymerization activity, as well as affecting the molecular weight and syndiotacticity [51].

Baird et al. used a cationic polymerization catalytic system in styrene syndiospecific polymerization [52]. They found that the reaction of Cp*TiMe_3_ combined with highly electrophilic tris(pentafluorophenyl)borane, B(C_6_F_5_)_3_, formed the methyl-bridged complex Cp*TiMe_2_(μ-Me)B(C_6_F_5_)_3_ (**28**), which initiated the cationic polymerization of styrene in toluene at room temperature and above in a syndiospecific manner, while at −15 °C and below, the system initiated styrene random polymerization in aromatic solvents to yield APS. In addition, the researchers found that styrene protonation produced a relatively stable cation, which was prone to cationic polymerization when the styrene ring was substituted with an electron-donating group.

Extended studies have shown that the addition of electron-donating substituents, such as methyl and higher alkyls, to the Cp ring can enhance the thermostability of the electrophilic metal centers, and bulkier substituents may also occupy the spatial volume of the coordination sphere as well as increasing the Lewis basicity of the active central metal, which in turn leads to a decrease in catalytic activity and stereoselectivity. Furthermore, the use of indenyl ligand in place of the substituted Cp’ ring can largely improve the activity, stereoselectivity, and thermal stability of the complex [53], and aryl-substituting Cp rings can effectively improve the catalytic activity and stereoselectivity of the mono-Cp’ titanocene complexes [54].

Ready et al. reported a mono-indenyl titanocene complex, IndTiCl_3_ (**29**) (Ind = η^5^-indenyl, Figure 3). Complex **29** exhibited considerably higher catalytic activity and syndioselectivity than the classic cyclopentadienyl titanocene trichloride complex, CpTiCl_3_ [53]. Under identical experimental conditions, SPS obtained from the mono-indenyl titanocene complex possessed over 90% syndiotacticity, compared with 60–80% from the simple mono-Cp’ titanocene complex.

Xu et al. reported a series of indenyl titanium trifluoride complexes, Ind’TiF_3_ (**30**–**38**) (Figure 3). The results showed that the catalytic activity and stereoselectivity of the fluorinated complexes were explicitly higher than those of the trichloride counterparts. Moreover, the catalytic activity and stereoselectivity of the mono-indenyl titanocene trifluoride complexes were prominently higher than those of the mono-Cp’ titanocene trifluoride homolog (Table 1).

In addition, Xu et al. also studied the effects of other substituents on mono-indenyl titanocene fluoride complexes. Only methyl-substituted complexes were explicitly more active than their unsubstituted counterparts, demonstrating that stronger electron-donating ability and smaller substituents could elevate the rate of chain growth while suppressing the rate of chain termination. Other large substituents increased the steric hindrance of the complex but interfered with styrene coordination and migration insertion. When the substituent was a phenyl group, however, the catalytic activity was higher than that of the unsubstituted analog. It was speculated that the electronic effect of the phenyl substituent was stronger than its steric effect in this case. On the other hand, the steric effect reduced the syndiotacticity of the SPS obtained with the phenyl-substituted mono-indenyl titanocene complex [55].

Qian et al. reported mono-alkoxy-substituted mono-indenyl titanocene complexes, IndTiCl_2_(OR) (R = Me, Et, Pr, Cy = cyclohexyl) (**39**–**42**) (Figure 3). Substitution of Cl atoms with one single alkoxy group could enhance the catalytic activity of the mono-Cp’ titanocenes [49,50,52]. For the mono-indenyl titanocene complexes, the catalytic activity could also be improved to a considerable magnitude by introducing a σ-coordinated ancillary alkoxy ligand. It was believed that the substitution of a halogen atom with a σ-coordinated alkoxy ligand introduced both electronic and steric effects. The electron-donating effect of the alkoxy group reduced the bonding energy of the remaining Ti–Cl bonds, resulting in easier alkylation and subsequent heterolytic cleavage of the Ti–C bond and cationization of the central metal. When a bulkier alkoxy group was introduced, such as the cyclohexyl group in complex **42** (Figure 3), the coordination and insertion of styrene in the cationic Ti center were hindered, resulting in reduced catalytic activity [56]. Schneider et al. reported phenanthrene-5,6-fused-cyclopentadienyl titanocene trichloride complexes (**43**–**45**) (Figure 3), and the styrene polymerization results showed that the complex with a phenyl substituent (**45**) exhibited the highest catalytic activity [57].

### 3.3. Mono-Cp’-RE Metal Catalysts

RE metals have a stable 3+ oxidation state, and RE cations generally have stronger Lewis acid and Lewis base (LB) affinity. RE–carbon bonds and RE–hydrogen bonds are relatively more active. Thus, the reactivity of the RE–C bonds in mono-Cp’RE active catalysts can be adjusted by modifying the η^5^-coordinated Cp’ ligand structure and/or changing the RE metal atoms in the active center. It has been reported that some dicyclopentadienyl RE metal complexes (Cp’_2_RE-R) can promote the polymerization of some conventional olefinic monomers without the assistance of a cocatalyst. However, these Cp’_2_RE-R complexes usually show negligible catalytic activity toward styrene and conjugated diene polymerization [58,59]. The mono-Cp’RE metal complexes, Cp’RE-R_2_, reportedly show much higher catalytic activity and diverse reactivity toward styrene polymerization.

In general, mono-Cp’RE-R_2_ complexes can be prepared by alkane elimination via the metathesis reaction between the trialkyl RE complexes, RE-R_3_, and an equivalent neutral Cp’-H ligand, or by the metathesis reaction between the RE trihalides (RE-X_3_; X = F, Cl, Br) and the corresponding ligand alkali metal salt (Cp’M; M = Li, Na, K). Such mono-Cp’RE-R_2_ metallocene complexes show high activity and stereoselectivity for the polymerization of α-olefins, dienes, styrene, etc. under the activation of high Lewis acidic boron salts [60].

Hou et al. reported Cp’RE-R_2_L-type complexes (C_5_Me_4_SiMe_3_)RE(CH_2_SiMe_3_)_2_(THF) (RE = Sc, Y, Gd, Lu) (Figure 4, **46**) [61]. At ambient temperature in toluene, complexes **46** did not exhibit catalytic activity toward olefins. However, when **46** were combined with one equivalent of [Ph_3_C][B(C_6_F_5_)_4_, the polymerization of styrene with high catalytic activity and syndiospecificity was achieved. As the ratio of styrene monomer to catalyst increased, the resulting SPS molecular weight increased almost linearly, and the SPS molecular weight distribution was narrow. The four reported (C_5_Me_4_SiMe_3_)RE(CH_2_SiMe_3_)_2_(THF) complexes showed highly efficient syndiospecific polymerization of styrene (rrrr > 99%), and among the four, the scandium complex (**46-Sc**) showed the highest catalytic activity (up to 13,618 kg mol^−1^h^−1^).

Okuda et al. reported mono-Cp’ dibenzyl scandium complexes (Figure 4, **47, 48**). When activated with [Ph_3_C][B(C_6_F_5_)_4_, complexes **46** and **47** exhibited catalytic activity toward styrene polymerization. Compared with **46**, complex **47** had relatively poor catalytic activity, and it was believed that the active metal center was more inclined to coordinate with the benzyl group and one benzyl ligand remained close to the metal center in an η^6^-coordinated manner, thus maintaining the stability of the metal center and resulting in the reduced catalytic activities of **46** and **47** [62]. Okuda’s group reported another series of mono-Cp’RE complexes, [Cp’RE = [η^5^-C_5_Me_4_{SiMe_2_(C_6_F_5_)]RE(CH_2_SiMe_3_)_2_(THF)] (RE = Sc, Y, Lu) (Figure 4, **49**), and dimetyl-2-furanylsilyl(tetramethylcyclopentadienyl) complexes with central metals Sc, Y, and Lu, [RE{η^5^-C_5_Me_4_SiMe_2_(C_4_H_2_O-2-R)}(CH_2_SiMe_3_)_2_(THF)] (RE = Sc, Lu, Y; R = H, Me) (Figure 4, **50**, **51**). The polymerization results showed that the catalytic activity of the complexes was related to the size of the central metal, which was manifested as (Sc > Y ≈ Lu). On the other hand, the dependence on the type of substituent R on the Cp ring was weaker [63,64]. It was also clarified that the appropriate amount of alkylaluminum reagent could improve the catalytic activity, while excessive Lewis acidic alkylaluminum suppressed the catalyzed polymerization kinetics.

In previous studies, active mono-Cp’RE complexes containing σ-coordinated ancillary alkyl ligands were often stabilized with additional LB, such as THF or amino groups. However, the strong coordinative nature of the LB hindered the olefinic monomers from coordinating with the active RE centers, often resulting in reduced reactivity. In later studies, it was found that stable mono-Cp’RE metal complexes without a coordinated LB could be prepared by increasing the steric bulk of cyclopentadienyl ligands or introducing chelated ligands, such as *o*-N,N-dimethyl benzyl, and allyl ligands in an η^3^ coordination mode [23].

Luo et al. reported a series of mono-Cp’-scandium bis(amido) complexes, C_5_Me_4_RSc[N(SiR’Me_2_)_2_]_2_·n[THF] (R = Me, SiMe_3_; R’ = H, n = 1; R’ = Me, n = 0) (Figure 4, **52**–**54**), with thermally stable properties. In general, the RE metal amido complexes had a wide range of structural characteristics and better thermal stability. The complexes did not show catalytic activity toward styrene polymerization without cocatalyst activation. In the presence of boron salts, all three complexes showed high catalytic activity and syndiospecific selectivity, with catalytic activity in the order of **52** > **53** > **54**. Increasing the ratio of styrene to scandium increased the molecular weight of SPS, while the molecular weight distribution remained nearly constant, proving that the catalytic system had controllable catalytic properties. The addition of alkylaluminum led to increased catalytic activity but decreased SPS molecular weight and resulted in broader molecular weight distribution, suggesting that an alkylaluminum-induced chain transfer reaction was taking place [65].

Luo et al. also reported RE complexes with bis-allyl coordination, (C_5_Me_4_SiMe_2_NC_4_H_8_)RE(η^3^-C_3_H_5_)_2_ (RE = Sc, Y, Lu) (Figure 4, **55**), and RE metal complexes coordinated by disilylamido ligand, (C_5_Me_4_SiMe_2_NC_4_H_8_)RE[N(SiHMe_2_)_2_]_2_ (RE = Sc, Y, Lu) (Figure 4, **56**). These complexes were thermally stable and had better solubility in common organic solvents. Complexes **55** and **56** alone showed silent activity toward styrene while being activated with [Ph_3_C][B(C_6_F_5_)_4_], whereas complexes **55-Sc**, **55-Y**, and **55-Lu** all showed catalytic activity toward styrene polymerization, while the disilylamido complexes **56** remained inactive, strongly suggesting that the type of σ-coordinated substituent played an important role in catalytic performance. The addition of AliBu_3_ to the complex **56**-boric salt system significantly improved the catalytic activity of the system. Therefore, the influences of the type of central metal and σ-coordinated substituent on the central metal on the polymerization reaction were obvious, and the scandium complexes and RE metal allylic complexes had higher catalytic activities [66].

Hou et al. reported mono-Cp’ RE metal complexes with *o*-N,N-dimethyl benzyl as σ-coordinated intramolecular LB chelating ligands (Figure 4, **57**, **58**) [67,68]. The phospheno-based complexes **57** showed limited catalytic activity toward styrene polymerization when activated with [Ph_3_C][B(C_6_F_5_)_4_], and it is worth noting that complex **57** still showed reasonable activity when the metal center was Y, while complex **57-Sm** was catalytically silent. To a certain extent, the experimental results showed that the larger the radius of the RE ions in the active center, the lower the catalytic activity toward styrene polymerization. Later, Hou’s group reported a class of RE metal complexes **58** with an η^5^-coordinated anionic pyrrolyl ring structure. The η^5^-coordinated pyrrolyl RE complexes also showed catalytic activity toward styrene polymerization when activated with [Ph_3_C][B(C_6_F_5_)_4_], and complex **57-Sc** showed catalytic activity of 3100 kg mol^−1^h^−1^. However, when the active centers were Y and La, the pyrrolyl complexes **58-Y** and **58-La** did not polymerize styrene, which showed that the catalytic activity of similar complexes was closely related to the radius of the RE cations in the active centers.

In recent years, mono-Cp’RE metallocene constrained geometry catalysts (CGCs) have shown excellent performance in styrene syndiospecific polymerization. Cui et al. reported a CGC-Lu complex, [(C_5_Me_4_C_5_H_4_N)Lu(η^3^-C_3_H_5_)_2_] (Figure 5, **59**), with an *o*-pyridine-substituted mono-Cp’ and bis-allyl coordinated structure [69]. Complex **59** had high activity and stereoselectivity for styrene syndiospecific polymerization. Activated with [Ph_3_C][B(C_6_F_5_)_4_] in toluene solution at room temperature, the system yielded near-perfect SPS (rrrr > 99%) microstructure. Furthermore, the molecular weight of SPS increased linearly with increasing monomer concentration, while the molecular weight distribution remained nearly constant (Mw/Mn = 1.88~1.98). The catalytic system of **59** could reach catalytic activity above 6240 kg mol^−1^h^−1^.

Subsequently, Cui et al. reported three CGC-RE metal complexes with different ligand structures (Figure 5, **60**–**62**) to study the CGC-ligand structural effects [70]. The three CGC-RE metal complexes all formed ternary catalytic systems with the activator [Ph_3_C][B(C_6_F_5_)_4_] and Al(iBu)_3_. The ternary systems showed activity in styrene polymerization at room temperature. The CGC-RE metal complexes **60** and **61** had low catalytic activity at room temperature, and the resulting SPS stereospecificity was not high (rrrr ≤ 84%). On the other hand, the CGC-RE metal complex **62** showed excellent activity. The CGC-RE metal complex of the Flu-CH_2_-Py (Flu = fluorenyl anion, Py = pyridinyl) catalytic system could reach up to 15,600 kg mol^−1^h^−1^ with near perfect syndiotactic (rrrr > 99%) microstructure. The Cui group reported the pyridine-fluorene CGC-RE complex [(Flu-*o*-C_5_H_4_N)Y(CH_2_SiMe_3_)_2_(THF)_n_] (Figure 5, **63**) [71]. Comparing **63** with previously reported CGC-RE metal catalytic systems, complex **62-Y** had higher activity and stereoselectivity, and it was believed that the fluorene η^3^ coordination mode of complex **63** inhibited electron delocalization in the large aromatic π-ring and hindered the fluorenyl π-ring from absorbing and delocalizing the electron density from the active RE metal center. DFT calculations for complexes **62**–**63** found that the reason for the lower activity and selectivity of complex **63** may have also originated from the stronger coordination of the THF molecule with the central metal, thus blocking the coordination of styrene monomer [72]. Cui et al. used the CGC-RE complex **62** catalyst structure, the fixed Flu-CH_2_-Py ligand framework, but changed the central metal with Sc, Y, Pr, Nd, Gd, Tb, Dy, Ho, Er, Tm, and Lu to study the central metal’s effects. The results showed that when the central metal was Sc, Y, Gd, Tb, Dy, Ho, Er, Tm, and Lu, the complexes showed high activity and selectivity, while when the center metal was Pr and Nd, the activity of the complex was largely reduced. This attempt showed that, for Flu-CH_2_-Py-RE complexes, the catalytic activity increased with the decrease in RE ion radius, indicating that the higher Lewis acidity (or higher ionic potential) in the central metal was conducive to the styrene coordination polymerization activity. It is worth noting that this result was contrary to previous research findings, and the researchers believed that different coordination polymerization mechanisms may have been in operation in this catalytic system. DFT calculations found that the lowest unoccupied orbital (LUMO) of the RE metal complex had a significant influence on the styrene polymerization activity. The degree of participation of the ligands in the LUMO orbital of the catalyst was related to the catalytic activity, and the ligand played an important role in reducing the LUMO energy of the active metal; thus, the reduced LUMO energy eased the ability of styrene to coordinate with the RE metal center, resulting in increased polymerization activity [73].

Further studying Ind-CH_2_-Py-RE metal complex **62** and Flu-CH_2_-Py-RE metal complex **63** revealed the effect of the cyclopentadiene segment structure on the activity of complexes [74]. Trimethylsilyl-substituted indenyl complex **64** (Figure 5) exhibited much less activity than complexes **62** and **63** with the Flu-CH_2_-Py-RE structure. Compared with the unsubstituted fluorenyl complex **63**, when electron-donating tert-butyl substituents were introduced at the remote 2,7 position of the fluorene moiety, the RE metal complex **65** (Figure 5) exhibited higher catalytic activity when the active metal was an element with a small atomic radius, such as Sc, Y, and Lu.

## 4. Summary

At present, SPS is widely used in automotive, construction, medical, and a variety of other applications, and it occupies an indispensable share of the world’s engineering plastics market. The use of metallocene catalytic systems can effectively produce SPS. In this text, mono-Cp’ metallocene catalysts, with active center metals involving the Group 4 element titanium and rare-earth (RE) metals, for styrene syndiospecific polymerization are introduced. Titanium and RE metals are widely used in the preparation of specific-purpose metallocene catalysts. In recent years, mono-Cp’-RE metal complexes have shown outstanding performance in the catalytic syndiospecific polymerization of styrene, and the ability to regulate polymerization activity and stereospecificity can be realized via tuning the ligand structure, selecting an appropriate Group 4 or RE metal atom, or through cocatalyst structure and activation processes.

After decades of development, mono-Cp’ metallocene complexes have made significant advances in the polymerization of olefins, dienes, and styrene, and some mono-Cp’ metallocene catalytic systems have been employed in industrial processes. However, the field of mono-Cp’ metallocene-catalyzed olefin and styrene polymerization is still full of potential. In exploring the polymerization mechanisms of mono-Cp’ metallocene catalytic systems with different structures, fine-tuning the ligand structural features, optimizing the cocatalyst components, and selecting appropriate central active metals are three effective ways to maximize the catalytic performance of mono-Cp’ titanocene and mono-Cp’-RE catalytic systems for the syndiospecific polymerization of styrene. Further exploring the polymerization processes and iterating the adoption of catalytic systems in industrial processes are still the main focuses of industry and academic researchers. The authors have no intention of providing a comprehensive review on SPS catalysis and hope this concise review on mono-Cp’Ti and mono-Cp’RE catalysts will provide a gateway for industry and academic researchers to follow up on the recent progresses in this rich and innovative field with high potential.

## Figures and Tables

**Figure 1 molecules-28-06792-f001:**
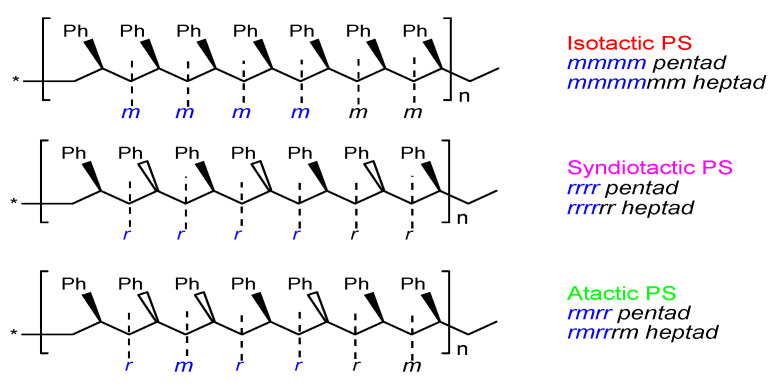
Fischer projection of three PS macromolecular chains and *meso (m)* vs. *racemic (r)* configurations. The star * at the chain ends represent repeating polymeric sigment of the same stereo-sequence.

**Figure 2 molecules-28-06792-f002:**
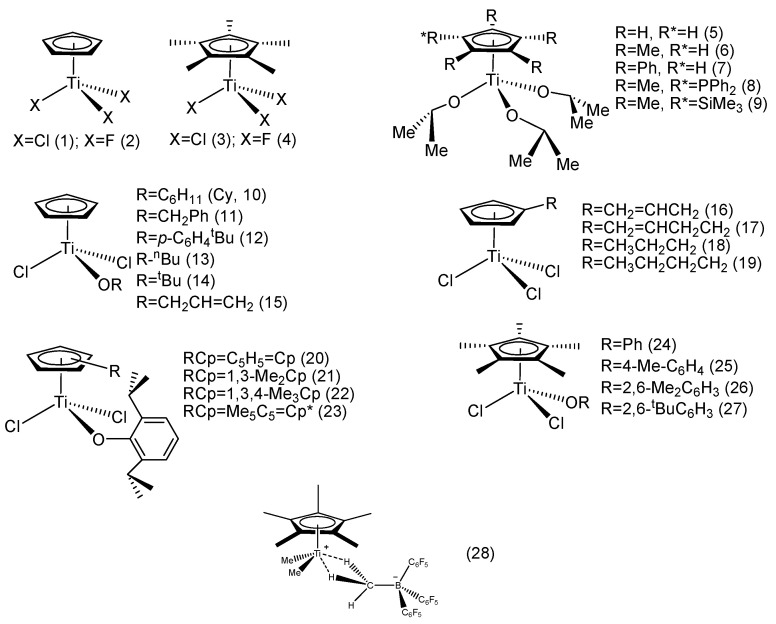
Mono-Cp’-TiX_3_ catalytic systems for styrene syndiospecific polymerization.

**Figure 3 molecules-28-06792-f003:**
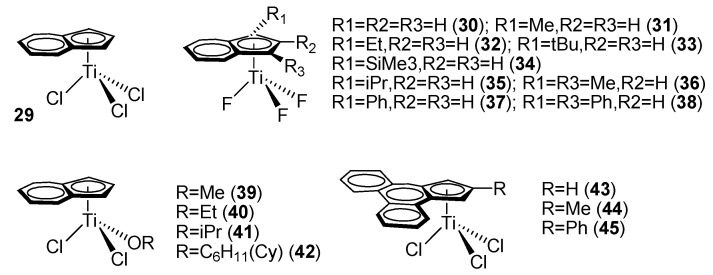
Ind’-TiX_3_ catalytic systems for styrene syndiospecific polymerization.

**Figure 4 molecules-28-06792-f004:**
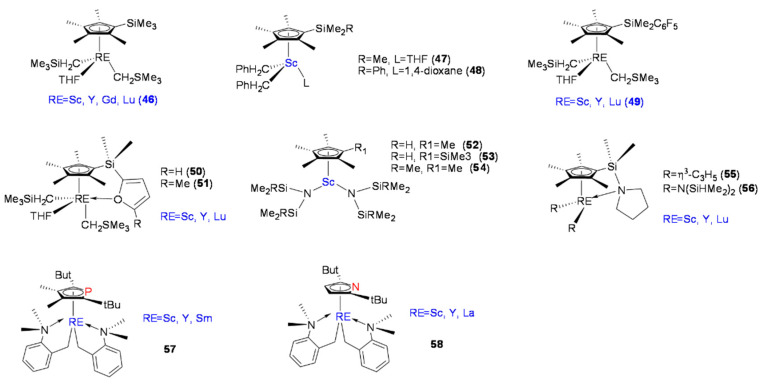
Mono-Cp’ RE complexes (**46**–**58**) for styrene syndiospecific polymerization.

**Figure 5 molecules-28-06792-f005:**
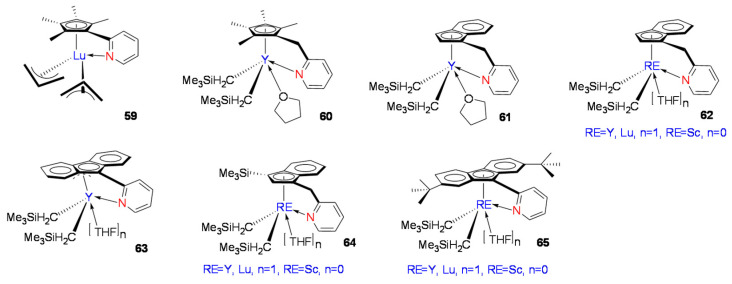
Mono-Cp’ RE complexes (**59**–**65**) for styrene syndiospecific polymerization.

**Table 1 molecules-28-06792-t001:** Selective metallocene-catalyzed styrene syndiospecific polymerization.

Complexes	*T_R_* (min)	*T_P_* (°C)	Al/Ti Ratio	Activity (kgPS/mol.M)	*T_melt_* (°C)	Mw (10^−5^ g/mol)	Syndiotactic Index (%)
CpTiCl_3_	10	50	300:1	198	258	0.5	75.3
CpTiF_3_	10	50	300:1	2800	265	1.2	86.2
IndTiCl_3_	10	50	300:1	400	268	1.0	94.3
IndTiF_3_	10	50	300:1	12,500	275	5.2	98.1

## Data Availability

All data in this review can be find in the references provided, together with the articles’ ISBN and/or DOI numbers. New data related to our research findings will be published elsewhere later.

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
