# Peer review of "High-Efficiency Mono-Cyclopentadienyl Titanium and Rare-Earth Metal Catalysts for the Production of Syndiotactic Polystyrene"

_molecules, 2023, doi:10.3390/molecules28196792_

Round 1
Reviewer 1 Report
In the paper High-efficiency Mono-Cyclopentadienyl Metal catalysts for syndiotactic polystyrene catalysts based on the mono-Cp' titanocene (mainly Cp'-Ti-(III) species) and the mono-Cp' Rare Elements (Cp'-RE) metallocenes for syndiospecific polymerization of styrene are discussed. There is not a correlation between the title and the content of the manuscript; the title refers to a broader transition metal catalysts while the authors discuss only Ti and Rare Elements. More results obtained with the corresponding catalysts, as well as important aspects on the reaction mechanism and stereochemistry should be included. The text and Fig. 3.3 and 3.4 displaying formulae with Re as transition metal are not correct. Re is the symbol for Rhenium whereas the authors treat catalysts based on Rare Elements (RE). The paper needs major revision, in the present form it is not acceptable for publication. The English style should be revised.
In the paper High-efficiency Mono-Cyclopentadienyl Metal catalysts for syndiotactic polystyrene catalysts based on the mono-Cp' titanocene (mainly Cp'-Ti-(III) species) and the mono-Cp' Rare Elements (Cp'-RE) metallocenes for syndiospecific polymerization of styrene are discussed. There is not a correlation between the title and the content of the manuscript; the title refers to a broader transition metal catalysts while the authors discuss only Ti and Rare Elements. More results obtained with the corresponding catalysts, as well as important aspects on the reaction mechanism and stereochemistry should be included. The text and Fig. 3.3 and 3.4 displaying formulae with Re as transition metal are not correct. Re is the symbol for Rhenium whereas the authors treat catalysts based on Rare Elements (RE). The paper needs major revision, in the present form it is not acceptable for publication. The English style should be revised.
Reviewer 2 Report
In the manuscript entitled: “High efficiency Mono Cyclopentadienyl Metal catalysts for syndiotactic polystyrene” the authors give a nice review of the catalytic activity of several cyclopentadienyl metal catalysts. The theme of the manuscript is interesting and gives a contribution to chemistry. Nevertheless, there are a couple of problems that need correction to improve the quality of the manuscript. I recommend the publication of this manuscript after the necessary corrections.
In the Abstract, 1st page, 19th row: Instead of IVB transition metals Group 4 transition metals should be used throughout the whole paper. The IUPAC recommendations should be consulted. What is RE? Is this abbreviation used for rare earth elements? If it is, then it should be indicated in the text at first time of its mentioning.
Rows 20-21: “In this context, the mono-Cp' titanocene (mainly Cp’-Ti-(III) species) catalysts, and the mono-Cp' RE metals ( Cp’-Re) metallocene catalysts…” Between Ti and (III) the hyphen should be deleted. The abbreviations RE and Re should not be mixed. Re is the symbol of rhenium and it is not a rare earth element.
Row 67: “differences of the phenyl on the linear aliphatic main chain”
Here most probably phenyl stands for the phenyl group on the main chain, so it should be written as “phenyl group” or “phenyl substituent”.
Row 85: a blank space is missing between “the 1950s”.
Rows 113-115: The abbreviations “mmmm” and “rrrr” should be defined. What is m and what is r?
Row 235: An o is missing from mon-Cp’.
Table 1: The quantities in the heather of the table should be defined.
Rows 323-326: Is LB affinity stands for Lewis base affinity? “Ln-C bonds” stands for lanthanoid – C bonds? The part of the sentence “the reactivity of the bonds can be adjusted by modifying ligands and changing the active center of RE ions.” is not completely clear. The RE ions can be the active centers of the catalysts, but they as single ions surrounded by coordinated cyclopentadienyl groups can’t have different active centers. The Cp-RE type catalysts as molecules may have different active centers.
Row 355: “mono-Cp’Re-“ should be corrected because this series does not contain rhenium (Re).
Row 375: “thermoly” should be replaced by “thermally”.
Row 399: Between the N, N-dimethyl the blank space should be deleted.
Round 2
Reviewer 1 Report
After examining the revised version of the manuscript, I accept publication in Molecules of the revised form
-
Reviewer 2 Report
I agree with the answers of the authors.
I have read the corrected manuscript and after the correction of a few small problems it can be accepted for publication. Please find my comments attached.
